# BIAS PROPAGATION IN FEDERATED LEARNING

**Hongyan Chang & Reza Shokri**
School of Computing
National University of Singapore
`{hongyan,reza}@comp.nus.edu.sg`

## ABSTRACT

We show that participating in federated learning can be detrimental to group fairness. In fact, the bias of a few parties against under-represented groups (identified by sensitive attributes such as gender or race) can propagate through the network to all the parties in the network. We analyze and explain bias propagation in federated learning on naturally partitioned real-world datasets. Our analysis reveals that biased parties unintentionally yet stealthily encode their bias in a small number of model parameters, and throughout the training, they steadily increase the dependence of the global model on sensitive attributes. What is important to highlight is that the experienced bias in federated learning is higher than what parties would otherwise encounter in centralized training with a model trained on the union of all their data. This indicates that the bias is due to the algorithm. Our work calls for auditing group fairness in federated learning and designing learning algorithms that are robust to bias propagation.

## 1 INTRODUCTION

Machine learning models can exhibit bias against demographic groups. Previous research has extensively studied how machine learning algorithms can reflect and amplify bias in training data, especially in centralized settings where data is held by a single party (Hardt et al., 2016; Dwork et al., 2012; Calders et al., 2009; Hashimoto et al., 2018; Zhang et al., 2020; Blum and Stangl, 2020; Lakkaraju et al., 2017). However, in practice, data is commonly owned by multiple parties and cannot be shared due to privacy concerns. Federated learning (FL) provides a promising solution by enabling parties to collaboratively learn a global model without sharing their data. In each round of FL, parties share their local model updates computed on their private datasets with a global server that aggregates them to update the global model. Despite the widespread adoption of FL in various applications such as healthcare, recruitment, and loan evaluation (Rieke et al., 2020; Yang et al., 2019), it is not yet fully understood how FL algorithms could magnify bias in training datasets.

Recent studies have investigated the problem of measuring and mitigating bias in federated learning with respect to a single global distribution (Chu et al., 2021; Zeng et al., 2021a; Hu et al., 2022; Du et al., 2021; Abay et al., 2020; Papadaki et al., 2021; 2022; Hu et al., 2022). However, in practice, parties often have *heterogeneous* data distributions. Evaluating the model's bias with respect to the global distribution does not accurately reflect the fairness of the FL model with respect to parties' local data distributions, which are relevant to end-users. This is the critical problem that we address in this paper. Specifically, we investigate the following questions: How does participating in FL affect the bias and fairness of the resulting models compared to models which are trained in a standalone setting? Does FL provide parties with the potential fairness benefits of centralized training on the union of their data? Can parties with biased datasets negatively impact the experienced fairness of other parties on their local distributions? How and why does the bias of a small number of parties affect the entire network? To the best of our knowledge, we provide the first comprehensive analysis of how FL algorithms impact local fairness.

We provide an empirical analysis based on real-world datasets. We show that FL might not sustain the benefits of collaboration in terms of fairness, as compared to its accuracy benefit. Specifically, compared with the standalone models, we find that the model trained in a centralized setting can be, on average, fairer on local data distributions. However, in those cases, the FL models, trained on the same dataset, can be more biased. This suggests that the FL algorithm itself can introduce new

bias in the final model. Furthermore, we demonstrate that FL impacts different parties in different ways. Specifically, we find a strong correlation between parties' fairness gap in the standalone setting and the fairness benefit they obtain from joining FL: parties with a greater bias in the standalone setting (caused by their local data) would receive a fairer model from FL. In contrast, FL has negative impacts on parties with a less significant bias in the standalone setting, resulting in a more biased model in FL. We further demonstrate that this is due to the fact that **FL propagates bias**: bias from a few parties can influence all parties in the network, hence aggravating the fairness problem globally.

Finally, we offer potential explanations for how bias is propagated in FL. Specifically, we show that *local updates* from biased parties increase the dependency between the model's predictions and the sensitive attributes. Such an increase is achieved by the norm increase in a small number of parameters (around $6\%$ of the model parameters in some experiments). This increase then propagates to the global model through aggregation, subsequently impacting all other parties. In addition, we show that the fairness gap of the final model can be governed by adjusting the value of those parameters. Surprisingly, we find that scaling these parameters can either reduce the model's bias to a small value of $0.05$ (on a measurement scale of 0 to 1) with only a $0.6\%$ drop in accuracy or increase the bias to a large value of $0.96$ with an $11\%$ drop in accuracy.

## 2 BACKGROUND

**Federated Learning.** In this paper, we consider the conventional federated learning (FL) setting. (McMahan et al., 2017). The FL framework consists of a network of $K$ parties, where each party $k \in [K]$ holds a local dataset $\tilde{D}_k$ of size $n_k$, sampled from a local data distribution $\mathcal{D}_k$. The objective of each party is to train a model that minimizes the loss on their local data distribution $\mathcal{D}_k$. To achieve this goal, FL trains a global model to minimize the average loss across parties, which is expressed as $\min_\theta \frac{1}{N} \sum_{k=1}^{K} n_k L(\theta, \tilde{D}_k)$, where $N$ is the sum of all local training dataset size. In each communication round $t$, a global server sends the current global model to all parties. All parties train the global model locally on their local dataset and send the updated local model to the server. The server then aggregates those local models to obtain the new global model. We consider the case where all parties participate in the training in each round, which helps to mitigate potential biases that could arise from the non-uniform sampling of participating parties.

**Group Fairness.** Fairness has a wide variety of meanings in literature. Group fairness entails, in particular, that the model should perform comparably across groups defined by sensitive attributes (e.g., sex). It is now common practice to evaluate discrimination in a model (or system) based on quantitative measurements of group fairness. In light of this, we focus on two widely-used group fairness notions, equalized odds (Hardt et al., 2016) and demographic parity (Dwork et al., 2012). To formally define those fairness notions, we assume each data point is associated with a sensitive attribute $a \in \mathcal{A}$, and we use $X$ and $Y$ to denote the input features and the true label. To measure fairness, we use the fairness gap with respect to Equalized Odds Difference, defined as

$$\Delta^{EO}(\theta, \mathcal{D}) := \max_{a, a' \in \mathcal{A}, y \in \mathcal{Y}} |\Pr_{\mathcal{D}}(f_\theta(X) = + | A = a, Y = y) - \Pr_{\mathcal{D}}(f_\theta(X) = + | A = a', Y = y)|$$

with an ideal value equal to zero (perfectly fair). Furthermore, in many applications, there exists a favorable prediction from the model (e.g., grant the loan). We assume the positive prediction $(+)$ as the favorable outcome. *Demographic parity* (Dwork et al., 2012) group fairness notion asks the model to give a favorable label to groups with equal rates. Similarly, the fairness gap with respect to Demographic Parity is defined as follows:

$$\Delta^{DP}(\theta, \mathcal{D}) := \max_{a, a' \in \mathcal{A}} |\Pr_{\mathcal{D}}(f_\theta(X) = + | A = a) - \Pr_{\mathcal{D}}(f_\theta(X) = + | A = a')|$$

with an ideal value equal to zero. In the rest of the paper, we use the fairness gap $\Delta$ (including $\Delta^{EO}$ and $\Delta^{DP}$ ) to measure the bias (fairness) of a model. The more significant fairness gap means the model is more biased (less fair). We will use bias or unfairness interchangeably.

**Measuring the impact of FL.** Federated learning aims to enhance model performance compared to standalone training and achieve comparable performance to centralized training. Thus, in order to evaluate the impact of FL on local fairness, we use centralized training and standalone training as baselines. In standalone training, each party trains a model $\theta_k$ independently to minimize the loss on its training data $\tilde{D}_k$. Note that the standalone model's fairness gap for a party is contributed by

herself. Therefore, we use the fairness gap of a party's standalone model to represent the party's bias. In contrast, in centralized training, all local training datasets from all parties are combined, and a centralized model $\theta_c$ is trained using this global training dataset. We define the benefit of FL and collaboration (i.e., centralized training) in terms of fairness and accuracy using these baselines. The benefit of FL for each party $k$ is defined as the difference between the standalone model $\theta_k$ and the FL model $\theta_g$. This difference reveals the extent to which a party benefits from participating in FL. Specifically, for a party $k$, we define the **accuracy benefit of FL** as $Acc(\theta_g, \mathcal{D}_k) - Acc(\theta_k, \mathcal{D}_k)$ and **fairness benefit of FL** as $\Delta(\theta_k, \mathcal{D}_k) - \Delta(\theta_g, \mathcal{D}_k)$. Similarly, the benefit of collaboration is based on the difference between the standalone model and the centralized model. This difference indicates how much an individual party could gain from collaborating with others by sharing the entire training dataset. We compute the average benefit of FL or collaboration across all parties in the network. A positive benefit indicates that FL (or collaboration) improves accuracy or fairness, while a negative benefit implies that FL or collaboration has a detrimental effect.

**Measuring bias aside fairness gap.** Measuring the bias of a model based on the fairness gap reveals the model's performance disparity across groups but does not explain why the model is biased. Specifically, it does not reveal whether the prediction of the model is attributed to the sensitive attribute or any other insensitive attributes whose distributions vary between groups. The former is typically referred to as disparate treatment (direct discrimination) as the sensitive attribute directly influences the model prediction (Grabowicz et al., 2022; Zafar et al., 2017). Towards measuring bias in this aspect, we employ the feature attribution method to explain the model's prediction on each data point. More precisely, given a model $f_\theta$, and the input features $x = (x_1, x_2, ..., x_d)$, the attribution of the prediction at input $x$ relative to a baseline input $x'$ is a vector $(a_1, ..., a_d)$ where $a_i$ is the contribution of $x_i$ to the prediction of $f_\theta(x)$. We use Integrated Gradient (Sundararajan et al., 2017) to compute the feature attribution, which only requires a few calls to the gradient operation. Following the suggestions by Sundararajan et al. (2017), we use the average feature values computed on the whole test dataset as the baseline for non-sensitive and non-binary sensitive features. For the binary sensitive attribute, we use the opposite feature value as the baseline. For instance, if the test input is (Sex: female, Age:28), the baseline input would be (Sex: male, Age: average age over the data points). By employing this feature attribution, we are able to discover the bias associated with disparate treatment. For more details about the Integrated Gradient, please refer to the original paper (Sundararajan et al., 2017).

## 3 EMPIRICAL ANALYSIS

Our objective is to understand how FL impacts fairness for parties. We start by comparing the average performance between FL and baselines (i.e., standalone training and centralized training) to answer the following question: ***does FL improve fairness for parties compared to standalone training, and does FL provide the same benefit as centralized training in terms of fairness?*** Our findings in Section 3.2 demonstrate that FL can exacerbate fairness issues for parties and does not retain the fairness benefit of collaboration, thus not achieving comparable performance to centralized training.

We further investigate how FL impacts performance for each party to answer the following questions: ***does FL provide the same benefit in terms of fairness for parties, and what causes the disparate impact of FL on parties' fairness?*** Our analysis in Section 3.3 shows that FL can propagate bias among parties. As a result, FL improves fairness for parties with greater bias while worsening fairness issues for parties with less bias.

Finally, we investigate ***how the bias is propagated in FL***. The analysis in Section 3.4 shows that biased parties encode the bias in a few parameters through local updates, which are propagated to the aggregated model and ultimately to other parties via parameter aggregation.

### 3.1 SETUP

We use the two datasets with different tasks for our empirical analysis: ***US Census Data*** and ***CelebA (Liu et al., 2015)***. The US Census dataset consists of census data from different places in the US. We naturally partition the dataset based on the source of the data. Thus, we have 51 parties in total, representing 50 states in the US and Puerto Rico. We consider three tasks defined

Table 1: **Average benefit of collaboration and FL**. The lowest accuracy and highest fairness gap are bold. The standard deviation across five runs is indicated between parentheses. The green arrows (red arrows) represent the positive (negative) impacts of FL or centralized learning compared to standalone training (i.e., FL or centralized training increases the accuracy or decreases the fairness gap compared to standalone training).

| Dataset | Accuracy | | | $\Delta^{EO}$ | | | $\Delta^{DP}$ | | |
|---|---|---|---|---|---|---|---|---|---|
| | Standalone | Centralized | FedAvg | Standalone | Centralized | FedAvg | Standalone | Centralized | FedAvg |
| Income- Race | **.738** (.001) | .787 (.003)↑ | .785 (.002)↑ | .506 (.026) | .478 (.013)↓ | **.514** (.021)↑ | .321 (.011) | .307 (.012)↓ | **.322** (.004)↑ |
| Income- Sex | **.738** (.001) | .787 (.003)↑ | .785 (.002)↑ | .146 (.007) | .177 (.030)↑ | **.198** (.007)↑ | .167 (.005) | .203 (.008)↑ | **.217** (.005)↑ |
| Health- Race | **.683** (.002) | .714 (.002)↑ | .709 (.001)↑ | **.226** (.013) | .17 (.021)↓ | .161 (.021)↓ | **.108** (.004) | .079 (.004)↓ | .072 (.008)↓ |
| Health- Sex | **.683** (.002) | .714 (.002)↑ | .709 (.001)↑ | **.042** (.003) | .034 (.002)↓ | .029 (.005)↓ | **.019** (.002) | .015 (.002)↓ | .013 (.004)↓ |
| Employment- Race | **.759** (.001) | .824 (.002)↑ | .823 (.001)↑ | **.374** (.023) | .243 (.010)↓ | .237 (.003)↓ | **.287** (.009) | .240 (.007)↓ | .241 (.010)↓ |
| Employment- Sex | **.759** (.001) | .824 (.002)↑ | .823 (.001)↑ | **.092** (.004) | .059 (.005)↓ | .059 (.003)↓ | **.054** (.003) | .040 (.007)↓ | .038 (.006)↓ |
| CelebA (Age)- Sex | **.812** (.004) | .852 (.007)↑ | .863 (.002)↑ | .219 (.01) | .222 (.024)↑ | **.264** (.012)↑ | .226 (.006) | .223 (.031)↓ | **.238** (.016)↑ |

in the folktables (Ding et al., 2021): Income (ACSIncome) [1], Health (ACSPublicCoverage), and Employment (ACSEmployment). We train two-layer neural network models for all the tasks. We consider the binary notion of sex (i.e., male and female) and multi-value race as the sensitive attribute for every task. Each party has $1,000$ training points and $2,000$ test points. Note that we use a large test dataset to ensure an accurate estimation of the fairness gap with respect to parities' test distribution. For CelebA, an image dataset consisting of $200,000$ celebrity images, we focus on the age prediction task and use the binary attribute "Male" (male and non-male) as the sensitive attribute, which we refer to as the Sex attribute. We partition the data uniformly at random among all parties and train CNN models. We also consider non-IID partitioning on the CelebA dataset, and the results are presented in Appendix B. In all experiments, we report the average results over five runs with different random seeds. We use FedML (He et al., 2020), a PyTorch (Paszke et al., 2019)-based library for FL, to train models in FL. We also use Captum (Kokhlikyan et al., 2020) to compute feature attribution. For further details on the datasets and models, please refer to Appendix A.

## 3.2 COLLABORATION VIA FL CAN WORSEN FAIRNESS ISSUE

Table 1 presents the average accuracy and fairness gaps of the centralized model, FL model, and standalone models on local datasets. Centralized training is observed to improve accuracy and fairness at the same time for parties. For instance, on the Income dataset, centralized training improves the accuracy by $6\%$ and reduces the fairness gap across racial groups by $5.5\%$ with respect to equalized odds and by $5.5\%$ with respect to demographic parity. However, FL may not achieve the same benefit as centralized training in terms of fairness and can even exacerbate the fairness issue for parties. For example, on the Income dataset, the EO fairness gap of the FL model for the sex groups increases by $35\%$, and the DP fairness gap increases by $29.9\%$, compared to standalone models. We include results for other popular FL algorithms, including FedNova (Wang et al., 2020), Scaffold Karimireddy et al. (2020b), FedOpt (Reddi et al., 2020), FedProx Li et al. (2020), and Mime (Karimireddy et al., 2020a), in Table 2 in Appendix B, which show a similar pattern to that of the FedAvg algorithm. Appendix B provides more detailed results on other FL algorithms. The centralized and FL models are trained on the same dataset. The difference between the FL model and the centralized model in terms of fairness suggests FL algorithm can introduce more bias compared to standard training. Therefore, the explanation of how bias is introduced during standard training in the centralized setting may not fully explain how FL introduces bias. In the following, we further explore how FL impacts fairness from the party level.

## 3.3 FL PROPAGATES BIAS AMONG PARTIES

**Disparate impact of FL on group fairness across parties.** Figure 1 illustrates the benefit of FL on fairness and accuracy for parties. We observe that FL improves accuracy for almost all parties, and the variance in accuracy improvement across parties is small. On the other hand, the fairness benefit of FL is negative for most parties. It implies that most parties obtain a more biased model in FL compared to standalone training. Furthermore, we notice that the variance in the fairness benefits across parties is large, suggesting that although all parties have the same global model in FL, they do not benefit from the FL model equally (each party evaluates the model performance on their local test dataset).

[1] We use the Adult Reconstruction dataset (Sarah et al., 2020).

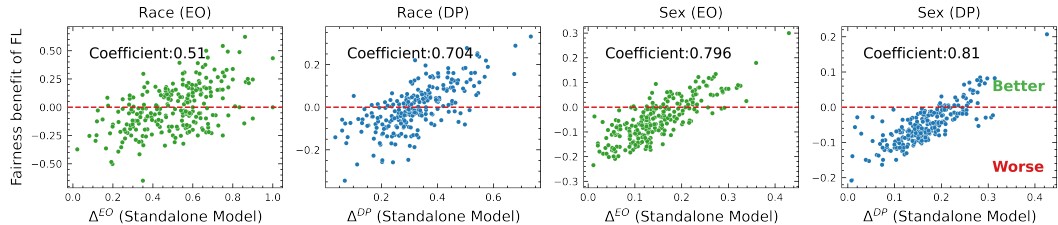

Figure 2: **Correlation between the fairness gap of the standalone model and the benefit of FL - Income.** The x-axis shows the fairness gap of the standalone model, and the y-axis shows the fairness benefit of FL, which is the fairness gap of the standalone model subtracted by that of the FL model (defined in Section 2). The Pearson correlation coefficients between the fairness gap of the parties' standalone models and the fairness benefit they obtain from FL are presented. The p-value for all settings is smaller than 0.0001.

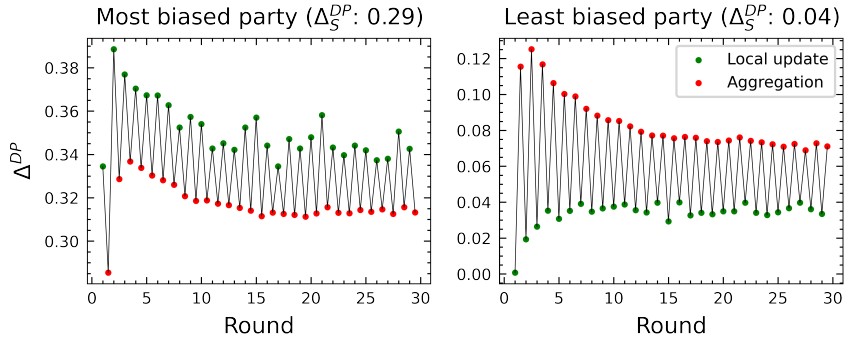

Figure 3: **Dynamic of fairness gap during the training - Income (Sex, DP).** Figure shows the fairness gap of the global model and locally updated model for the most biased party and least biased party in the first 30 rounds (Figure 16 in Appendix B shows the results for all training rounds.). The most (least) biased party has the highest (lowest) fairness gap in the standalone setting. The fairness gap in the standalone setting for those parties is shown in the title.

To explore the impact of FL on fairness at the party level, Figure 2 shows strong correlations between the fairness benefit a party obtains in FL and the fairness gap of the standalone model for the party, i.e., the bias level of the party. This finding highlights the disparate impact of FL on fairness: FL can improve fairness for more biased parties but at the cost of worsening the issue for less biased parties.

**Contradiction between aggregation and local update.** During the training process of FL, we observe that aggregation and local update contradict each other, shown in Figure 3. Local update from the least biased party, whose standalone model has the lowest fairness gap, reduces the fairness gap of the model. However, this reduction is eliminated by the aggregation step. Conversely, the local update from the most biased party increases the fairness gap of the model, which is then reduced by aggregation. This

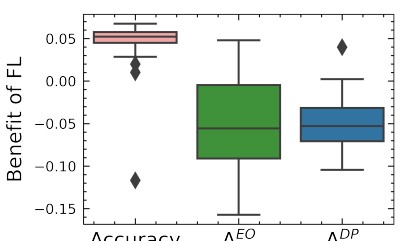

Figure 1: **Accuracy and Fairness Benefit of FL - Income (Sex).** The benefit of FL is the increase in accuracy or reduction in the fairness gap of the FL model compared to standalone training.

finding implies that the aggregation contributes to the disparate impact of FL on fairness, improving fairness for more biased parties but worsening the fairness for less biased parties compared to the standalone setting.

**Biased parties negatively influence other parties via aggregation throughout the training.** Why does aggregation have disparate impacts on local fairness for parties? Specifically, we ask which party's local update causes the increase (or decrease) of the fairness gap for other parties via

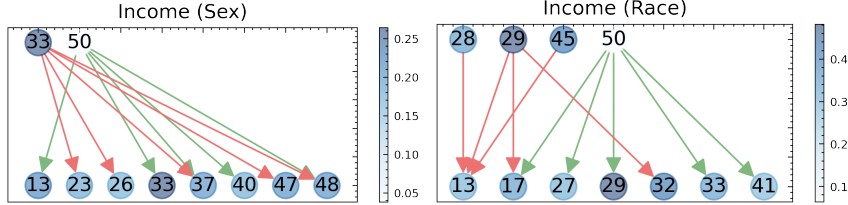

Figure 4: **Influence graphs - Income (DP).** The figure shows the most influential pairs of parties (with respect to demographic parity), with nodes on the top representing parties that influence other parties and the nodes on the bottom representing parties that are influenced by others. The number inside the node represents the party index, and the color of the node represents the fairness gap in the standalone setting. Green edges (resp. red edges) connect pairs of parties where the top party positively (negatively) influences the bottom party. We show the top 5 pairs with maximal positive influence and the top 5 pairs with maximal negative influence.

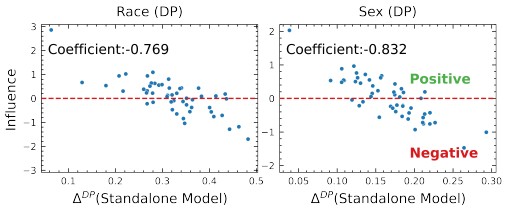

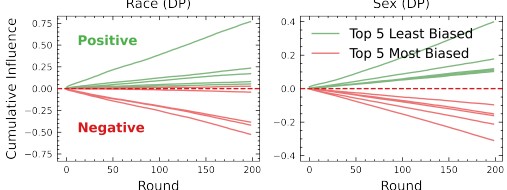

Figure 5: **Correlation between the influence and standalone bias - Income (DP).** The y-axis represents the average influence of each party. The Pearson correlation coefficient between the fairness gap a party obtains in the standalone setting and the impact she has on local fairness for parties in FL are shown in the figure.

Figure 6: **Cumulative influence - Income (DP).** The y-axis shows the cumulative influence of each party on all parties up until the current round. The results for the top five most biased parties (parties with the highest fairness gap) and the top five least biased parties (parties with the lowest fairness gap) are presented.

aggregation. To answer this question, we look into the influence of a party's local update on other parties' fairness via aggregation. More precisely, we compute the influence of party $i$ on party $j$ as the fairness gap increase when party $i$'s local update is removed from the aggregation in each round $t$ and sum over all the training rounds. Formally, we define the influence of party $i$ on party $j$ as $I_{i,j} = \sum_{t=1}^{T} \Delta(\theta_{t,-i}, D_j) - \Delta(\theta_t, D_j)$, where $\theta_t$ is the global model (i.e., the aggregated model overall local updated models) and $\theta_{t,-i}$ is the aggregated model over local updated models from parties excluding party $i$. If party $i$ improves fairness for party $j$, the influence is positive and vice versa. We compute the influence for all pairs of parties, and the most influential pairs are shown in Figure 4. We observe that a less biased party has a positive influence on fairness for other parties, while a more biased party has a negative influence on other parties. This result shows that a biased party can negatively influence other parties' fairness via aggregation throughout the training.

Furthermore, we investigate the relationship between a party's bias (i.e., the bias of the party's standalone model) and its average influence on all parties' fairness (i.e., $\sum_{k=1}^{K} I_{i,k}/K$). The results are presented in Figure 5, which shows a strong correlation between these two factors. This finding further supports our conclusion that the more biased parties have a stronger negative influence on other parties' fairness, while the less biased parties have a stronger positive influence.

Finally, we analyze the dynamics of the influence of the top 5 most biased parties (i.e., parties with the largest fairness gap in the standalone setting) and the top 5 least biased parties, as shown in Figure 6. We observe that the influence of the most biased parties is monotonically increasing throughout the training, while the influence of the least biased parties is decreasing. In other words, biased parties consistently have a negative influence on others' fairness gaps, while less biased parties have a positive influence. These results suggest that FL can propagate bias among parties: the bias from biased parties negatively influence the fairness of other parties via aggregation throughout the training. Next, we will explore how the bias is propagated in FL.

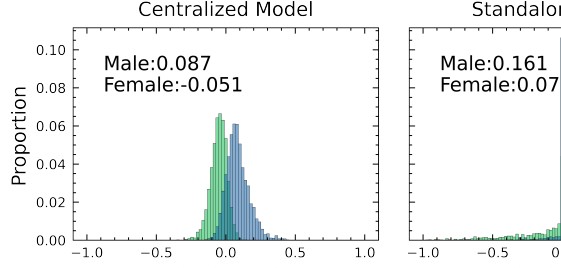
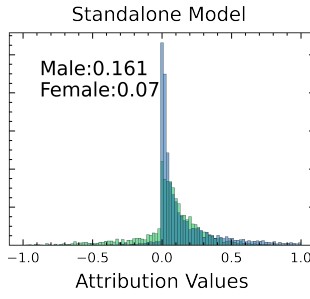
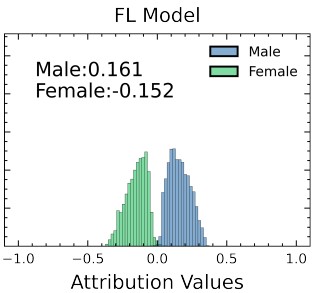

Figure 7: **Histogram of feature attribution value for the sensitive attribute - Income (Sex).** The average attribution value for the female group and male group is shown in the figure. The results are computed on all test data points over five different runs. We show the results for the most biased party with the highest fairness gap in the standalone setting.

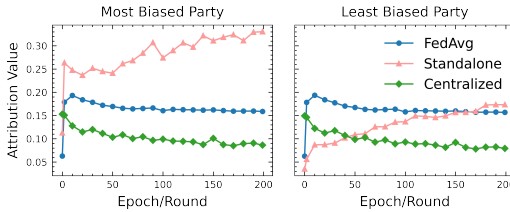
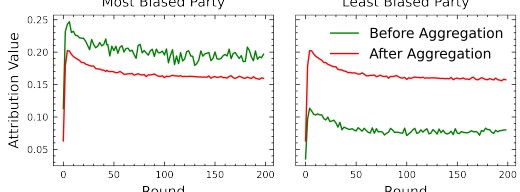

Figure 8: **Dynamic of absolute attribution value for the sensitive attribute - Income (Sex).** The results of different models are shown in different lines. We present the results for the most biased party and the least biased party.

Figure 9: **Effect of local update and aggregation on the attribution value - Income (Sex).** We show the attribution value for sex attribute before and after aggregation for the most biased party and the least biased party during the training.

### 3.4    How is bias propagated in FL?

**Disparate treatment causes large fairness gaps.** Our first investigation explores what the bias represents, specifically whether the bias increase in FL is directly caused by the disparate treatment of the model among sensitive groups. To answer this question, we utilize Integrated Gradients (Sundararajan et al., 2017) to measure the attribution of each input feature to the models' predictions with respect to the positive class. Figure 7 shows the attribution value distribution for the sensitive attribute "Sex" over individual test points from the female and male groups. We notice that the sex attribute has a large attribution value for the standalone model's predictions and the FL model's predictions. This finding implies that the predictions of those models are heavily dependent on the sensitive attribute of the test data. Moreover, the sensitive attribute affects the model predictions differently for the male and female groups, with the average attribution value being positive for the male group and negative for the female group. Furthermore, we find that there is minimal difference in the attribution value for other attributes with respect to the female and male groups (see Figure 19 in Appendix B). This indicates that the large fairness gap in the models is not caused by the distinct distribution of other insensitive attributes over protected groups; rather, it is mainly caused by the models' disparate treatment of protected groups.

**FL model learns more biased patterns.** In Figure 7, we compare the attribution value of "Sex" for different models and find that, while the predictions of the FL model depend less heavily on the sensitive attribute compared to those of the standalone model, the dependence is still stronger than that of the centralized model. This suggests that the FL model learns a more biased pattern compared to what could be learned in the centralized setting. Figure 8 shows the dynamic of the average absolute feature attribution for "Sex" during the training. We find that standalone training increases the model's dependence on the sensitive attribute throughout training for the most biased party, but centralized training decreases it. This suggests that collaboration through centralized training improves local fairness by guiding the model to learn less biased features. In FL, however, the absolute attribution value barely changes after 100 rounds and remains significantly greater than

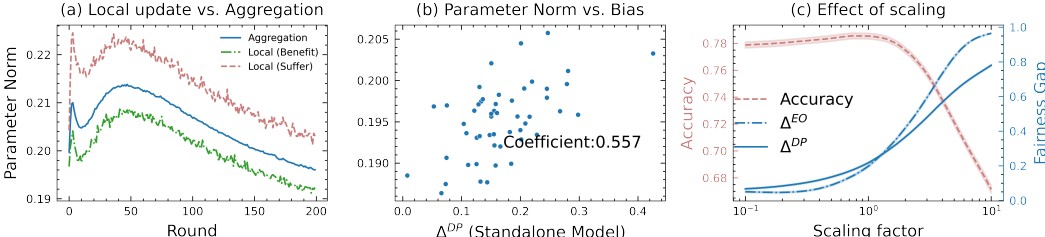

Figure 10: **Effect of a few parameters on fairness - Income (Sex).** *(a) Effect of local update and aggregation on the parameter values:* The y-axis shows the norm of parameters that are directly computed on the sensitive attribute, normalized by the parameter norm of the first layer. We show the dynamic of this parameter norm during FL for the aggregated model and locally updated model for the parties who benefit most from FL and suffer the most from FL (represented by the line with "Benefit" and "Suffer" respectively). *(b) Correlation between standalone bias and parameter values:* The x-axis shows the bias a party receives in the standalone setting, and the y-axis shows the normalized norm of parameters (associated with sex attribute) for the locally updated model in the last round. *(c) Effect of scaling the parameter values:* The figure shows the model performance in terms of accuracy and fairness gap when the parameter values (associated with sex attribute) are multiplied by a scaling factor.

that in the centralized setting. This implies that the FL algorithm may introduce bias to the final model by inhibiting the model from learning less biased features.

**Biased parties increase the model dependence on the sensitive attribute.** Figure 9 shows the attribution value of the aggregated model and locally updated model from the most biased party and least biased party. We observe that the biased party increases the global model dependence on sensitive attributes during the local update, and this increase persists throughout the training. In contrast, the least biased party reduces the dependence on the sensitive attribute, which is aligned with the trend of the fairness gap in Figure 3. This suggests that the biased parties have a negative impact on the fairness of other parties by increasing the model's dependence on the sensitive attribute.

**Bias is encoded in a few parameters.** Biased parties increase the model dependence on the sensitive attribute. But how does this increase propagate to other parties in the network? Since parties share the model parameters of the local model with the server, the bias is likely encoded in the model parameters. Therefore, we investigate which parameters are related to the model's bias. Intuitively, the parameters used to extract sensitive attribute information impact the attribution value of the sensitive attribute to the model prediction. If the absolute value (signal) of those parameters is substantial, the value of the sensitive attribute will significantly affect the model's prediction. In our evaluation, the sensitive attribute is part of the input feature, so the parameters directly applied to the "Sex" attribute in the first layer of the neural network should contribute to the model's bias.

Figure 10(a) shows the normalized norm of the parameters associated with the sensitive attribute for the most and least biased parties in the aggregated and locally updated models. The normalized norm is defined as the norm of parameters associated with the sensitive attribute divided by the parameter norm of the first layer. We observe that the least biased party reduces the parameter norm, hence decreasing the model's sensitivity to the sensitive attribute. On the other hand, the biased party increases the norm for those parameters, amplifying the impact of the sensitive attribute on the model's prediction. Through aggregation, this amplification will be propagated to the global model. Figure 10(b) reveals a moderate correlation between the fairness gap party experiences in the standalone setting and the normalized parameter norm for the parameters used to extract sensitive attribute information. This implies that biased parties increase the parameter value associated with sensitive attributes during the local update, thereby boosting the model's susceptibility to the sensitive attribute.

**Controlling fairness gap by scaling a few parameters.** To further investigate the impact of the parameters associated with the sensitive attribute (i.e., $104$ parameters out of $1,792$) on model fairness, we examine the effect of scaling these parameters on the fairness gap in Figure 10(c). We find that the fairness gap can be greatly widened or narrowed at the expense of a moderate degree of

accuracy. Specifically, by scaling the parameter value by $0.1$ for the trained FL model, we significantly reduce the EO gap by $74.6\%$ (from $0.198$ to $0.05$) and the DP gap by $69.1\%$ (from $0.217$ to $0.067$) with just a moderate accuracy loss of $0.8\%$ (from $0.785$ to $0.779$). In contrast, scaling the same set of parameters by a factor of $10$ increases the EO gap to $0.96$ (the maximal fairness gap is $1$), which is almost five times larger, and increases the DP gap by $259\%$ to $0.78$, while reducing the accuracy from $0.785$ to $0.67$. These findings explain how bias is propagated in FL: biased parties magnify the impact of sensitive attributes on model predictions by increasing the model parameter used to extract sensitive attributes. This rise in parameters is subsequently propagated to the global model through aggregation, further aggravating the issue of fairness for other parties. Our results explain how bias is propagated in FL: **Biased parties encode bias in a few parameters through a local update, and this bias is consequently propagated to the entire network through parameter aggregation.**

## 4 RELATED WORK

Fairness has received considerable attention due to the growing deployment of machine learning in decision-making processes. Various definitions of fairness have been presented (Hardt et al., 2016; Dwork et al., 2012; Calders et al., 2009). Specifically, group fairness requires that the model behave similarly for groups defined by a sensitive attribute (e.g., race). While how machine learning algorithms propagate data bias to the final model has been extensively investigated in a centralized setting (Blum and Stangl, 2020; Lakkaraju et al., 2017; Rambachan and Roth, 2020; Friedler et al., 2019; Dullerud et al., 2022), the effect of FL on model fairness is not yet fully understood.

The existing literature on fairness in FL mainly focuses on the performance disparity of FL models across parties, rather than demographic groups (Li et al., 2021; Zhao and Joshi, 2022; Li et al., 2019; Mohri et al., 2019; Deng et al., 2020; Donahue and Kleinberg, 2021; Hao et al., 2021; Zhou et al., 2021; Yu et al., 2020; Lyu et al., 2020). However, we focus on group fairness, which concerns performance disparity among groups. In terms of group fairness, Abay et al. (2020) listed a few potential sources of bias in FL. Recently, considerable progress has been made in training group fair models in FL (Abay et al., 2020; Chu et al., 2021; Zeng et al., 2021a; Hu et al., 2022; Du et al., 2021; Ezzeldin et al., 2021). Nonetheless, the majority of these works suggest techniques for achieving fairness on a single test distribution. Instead, we focus on fairness issues for parties. Some studies (Cui et al., 2021; Papadaki et al., 2022) proposed algorithms to improve local fairness for parties. Our purpose, instead, is to gain a comprehensive understanding of how FL influences local fairness on its own, which we believe is equally crucial as designing fair algorithms.

## 5 FUTURE WORK & CONCLUSION

**Future Work.** In this work, we have investigated how bias is propagated in FL when the sensitive attribute is included as an input feature. In practice, however, sensitive attributes may be prohibited from being included in input features. In such situations, the model may still be heavily biased due to variables that are correlated with the (unobserved) sensitive attribute. For instance, a person's zip code may be highly correlated with their race, a phenomenon known as "redlining". A promising direction for future work is to identify which model parameters contribute to the bias and to audit the bias propagation in this setting. Another important direction is to design FL algorithms that are robust to bias propagation. In Appendix E, we briefly discuss a few potential ways to achieve this goal.

**Conclusion.** Federated learning has become increasingly popular in various applications with significant individual-level consequences, making it essential to anticipate the possible bias introduced by FL. Our paper takes the first step in this direction by providing a comprehensive analysis of the impact of FL on local fairness for parties. We demonstrated that the FL algorithm could introduce bias on its own which may exacerbate the issue of fairness for the involved parties. Moreover, we showed that this exacerbation is not evenly distributed among parties, as FL can propagate bias among them. Finally, we explained how bias is propagated in FL: biased parties encode their bias into the local updates by increasing the signal of a few parameters steadily throughout the training process, which is then propagated to the global model via aggregation and, ultimately, to other parties.

## 6    ACKNOWLEDGEMENT

The authors would like to thank Ergute Bao, Ta Duy Nguyen, and Martin Strobel for their valuable feedback on earlier versions of this paper, as well as the anonymous reviewers for their insightful comments. This research is supported by Google PDPO faculty research award, Intel within the www.private-ai.org center, Meta faculty research award, the NUS Early Career Research Award (NUS ECRA award num- ber NUS ECRA FY19 P16), and the National Research Foundation, Singapore under its Strategic Capability Research Centres Funding Initiative. Any opinions, findings, conclusions, or recommendations expressed in this material are those of the author(s) and do not reflect the views of the National Research Foundation, Singapore.

## 7    ETHICS STATEMENT

Our analysis focuses on group fairness, which is typically used to audit the model or system for any bias or discrimination. Auditing the bias with respect to other definitions of fairness, such as individual fairness, may result in different conclusions. Moreover, our study focuses primarily on the binary notion of sex attributes and the multi-valued race attribute. We recognize that there are numerous protected groups outside those considered in the analysis, such as those defined by multiple sensitive attributes. The propagation of bias against fine-grained subgroups may be even more substantial than we found in the paper.

## 8    REPRODUCIBILITY STATEMENT

We provide details about the model, datasets, and implementations in Appendix A, and the code for the paper is available at `https://github.com/privacytrustlab/bias_in_FL`.

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

APPENDIX

This appendix is divided into five sections. Appendix A provides additional details about the experimental setups, including information about the models, datasets, and hyperparameters used. Appendix B presents further experimental results that support the claims made in the paper. In Appendix C, we discuss the results of an existing fair FL algorithm. Appendix D extends our analysis to real-world medical datasets. Finally, in Appendix E, we discuss potential methods for mitigating bias in FL.

## A  DETAILS ABOUT EXPERIMENT SETUP

### A.1  DATASETS AND MODELS

We explain the datasets and models used in the paper.

**Census Dataset**   We use the datasets provided by folktables. In particular, we consider the ACSIncome, ACSPublicCoverage, and ACSEmployment tasks defined in the forlktables. In the ACSIncome, the goal is to predict whether an individual's Income is above $50,000. In the ACSEmployment task, the goal is to predict whether an individual is employed. Similarly, in the Health (i.e., ACSPublicCoverage) task, the objective is to predict whether an individual is covered by public health insurance. We use the same pre-processing as in the folktables and train a fully connected neural network model with one hidden layer of 32 neurons for Income and 64 neurons for Employment and Health tasks. For all the tasks, we use the RELU activation function. We use an SGD optimizer with a learning rate of 0.001 for centralized training on Health and Employment datasets and 0.1 for other settings, and the batch size is 32. We train the NN models for 200 epochs. In FL, each client updates the global mdoel for 1 epoch and shares it with the server. We encode the categorical features based on the encoding template provided in folktables. After the encoding, the input feature size for Income is 54, 154 for Health, and 109 for Employment. We consider sex and race as sensitive attributes. Accordingly, there are two gender groups (male and female) and nine racial groups ("White alone," "Black or African American alone," "American Indian alone," "Alaska Native alone," and "American Indian and Alaska Native tribes specified; or American Indian or Alaska Native, not specified and no other," "Asian alone," "Native Hawaiian and Other Pacific Islander alone," "Some Other Race alone," "Two or More Races").

**CelebA Dataset**   We train CNN models on the dataset with one CNN layer whose output channel is 32, kernel size is 3, and stride is 1. We use 'same' padding for the CNN layer. Following this CNN layer, we have the Batch normalization layer and Max Pooling layer. After which, we have the connected layer. We train the model for 500 communication rounds or epochs with SGD optimizer. The learning rate is 0.1, and the batch size is 128. The train, test, and validation datasets ratio for each party is 6:2:2.

### A.2  HYPER-PARAMETER FOR FL ALGORITHMS AND IMPLEMENTATION

In our paper, we also evaluate various popular FL algorithms, including FedNova (Wang et al., 2020), Scaffold Karimireddy et al. (2020b), FedOpt (Reddi et al., 2020), FedProx Li et al. (2020), and Mime (Karimireddy et al., 2020a). We use the same local learning rate and the number of the local epoch as in FedAvg. We provided the detailed hyper-parameters for each of the algorithms in our code (See supplementary). We run all experiments on Ubuntu with two NVIDIA TITAN RTX GPUs.

### A.3  DATA HETEROGENEITY

Figure 11 shows the fraction of samples for each subgroup across all parties, while Figure 12 shows the histogram of subgroup proportions for each party. We observe that the parties have different fractions of samples from each group in the Income dataset, indicating that their local data distributions are dissimilar. In contrast, for the Health and Employment datasets, parties have similar fractions of samples from each subgroup, suggesting that their data distributions are more alike than those of the Income dataset.

# B   ADDITIONAL EXPERIMENTAL RESULTS

## B.1   BIAS PROPAGATION EFFECT

**Census dataset**   We show the bias propagation effect of FL for the Health and Employment task in Figure 13 and 14 respectively.

**CelebA**   We show the bias propagation effect of FL on CelebA dataset. Besides partitioning the data in an IID manner (we refer to as setting (i)), we also evaluate two different settings, where we change the number of samples from the minority subgroup (the female group with the "Not Young" age label). In this way, we aim to change the data bias in the local training datasets for clients. In particular, in setting (ii), half of the parties have more samples from the minority subgroup than another half of the parties. The ratio is 8:2. In setting (iii), a single party has half of the data from the minority subgroup, and the data is then iid partitioned among the other parties. In the figure, we find that the in the non-iid setting, the more biased parties have a larger and more positive benefit, while FL hurts the less biased parties. Figure 15 shows the correlation between the fairness gap of the standalone model and the benefit a party gets in the FL. We find that the in the non-iid setting, the more biased parties have a larger and more positive benefit, while FL hurts the less biased parties.

## B.2   AGGREGATION CONTRADICTS WITH LOCAL UPDATE

Figure 16 shows the dynamic of the fairness gap for the aggregated model and locally updated model from the most biased party and least biased party during the training.

## B.3   INFLUENCE SUB-GRAPHS

We show the top 10 maximal positive and top 10 maximal native influence client pairs in Figure 17 and the top 15 in Figure 18. We can see that the less biased client has a strong positive influence on other clients while the more biased client has a strong negative influence on other clients.

## B.4   ATTRIBUTION VALUE FOR SENSITIVE ATTRIBUTE

We show the attribution value for all input features for the most biased party and least biased party in Figure 19 and Figure 20, respectively. We observe that the attribution value for other features is similar for female and male groups. However, the sex attribute has the largest attribution value and affects the groups differently. It implies that the models are highly dependent on the sensitive attribute (i.e., "Sex") for making predictions.

## B.5   EFFECT OF FL ALGORITHM

Table 2 shows results on other FL algorithms, including FedNova (Wang et al., 2020), Scaffold Karimireddy et al. (2020b), FedOpt (Asad et al., 2020), FedProx Li et al. (2020), and Mime (Karimireddy et al., 2020a). We find that when the FL model achieves higher accuracy than standalone models, the fairness gap of the FL model can be higher than that of standalone models. This observation is consistent across multiple FL algorithms. This strong evidence implies that the improvement of accuracy in FL can come at the cost of fairness. In addition, this is not unique to only the FedAvg algorithm.

## B.6   GROUP PERFORMANCE

In Figure 21, we present the model ROC on groups to illustrate the performance disparity across groups. Surprisingly, there is no huge difference between the ROC between groups. In contrast, Figure 22 shows the noticeable difference between groups with respect to the precision-recall curve, especially for the most biased party. The main reason is that the dataset is imbalanced. Thus, ROC is usually misleading. On the precision-recall curve, we find that the model achieves a higher precision on the majority (i.e., Male group) compared to the minority group (i.e., Female group).

Table 2: **Benefit of collaboration on local datasets - Various FL algorithms (Income).** The standard deviation across five runs is indicated between parentheses.

| Algorithm | Accuracy | Race | | Sex | |
|---|---|---|---|---|---|
| | | $\Delta^{EO}$ | $\Delta^{DP}$ | $\Delta^{EO}$ | $\Delta^{DP}$ |
| Standalone | 0.738 (0.001) | 0.506 (0.026) | 0.321 (0.011) | 0.146 (0.007) | 0.167 (0.005) |
| FedAvg | 0.785 (0.002) | 0.514 (0.021) | 0.322 (0.004) | 0.198 (0.007) | 0.217 (0.005) |
| FedNova | 0.785 (0.002) | 0.517 (0.023) | 0.322 (0.004) | 0.203 (0.008) | 0.22 (0.006) |
| Scaffold | 0.695 (0.007) | 0.406 (0.213) | 0.233 (0.156) | 0.099 (0.047) | 0.077 (0.025) |
| FedOpt | 0.78 (0.001) | 0.525 (0.023) | 0.322 (0.008) | 0.201 (0.005) | 0.218 (0.004) |
| FedProx | 0.783 (0.002) | 0.515 (0.024) | 0.324 (0.006) | 0.199 (0.011) | 0.219 (0.009) |
| Mime | 0.786 (0.002) | 0.521 (0.027) | 0.321 (0.01) | 0.2 (0.01) | 0.218 (0.007) |

## B.7 EFFECT OF SCALING OTHER PARAMETERS

Figure 23 shows a comparison of the model's performance when scaling different model parameters. Specifically, we examine the impact of scaling the parameters in the first layer of the neural network that does not have any computation on the Sex attribute (referred to as "Other parameters"), as well as the parameter related to the Sex attribute (referred to as "Related parameters"). The results indicate that up-scaling or down-scaling other parameters do not significantly affect the fairness gap while scaling the related parameters has a considerable impact on model fairness. This finding supports our hypothesis that the model's bias is primarily encoded in a few model parameters. In FL, biased parties introduce bias during local training by increasing the weights for those related parameters.

Table 3: **Effect of Fair FL (Abay et al., 2020) - (Income).** The standard deviation across five runs is indicated between parentheses. The "Sensitive Attribute" column indicates the sensitive attribute used in the reweighting algorithm.

| Algorithm | Sensitive Attribute | Accuracy | Race $\Delta^{EO}$ | Race $\Delta^{DP}$ | Sex $\Delta^{EO}$ | Sex $\Delta^{DP}$ |
|---|---|---|---|---|---|---|
| Standalone | - | 0.738 (0.026) | 0.506 (0.19) | 0.321 (0.117) | 0.146 (0.071) | 0.167 (0.063) |
| FedAvg | - | 0.785 (0.017) | 0.514 (0.195) | 0.322 (0.086) | 0.198 (0.05) | 0.217 (0.037) |
| Global reweighting | Race | 0.783 (0.002) | 0.353 (0.016) | 0.212 (0.012) | 0.204 (0.006) | 0.219 (0.005) |
| Local reweighting | Race | 0.783 (0.002) | 0.367 (0.008) | 0.222 (0.005) | 0.196 (0.005) | 0.214 (0.006) |
| Global reweighting | Sex | 0.781 (0.001) | 0.516 (0.031) | 0.318 (0.01) | 0.044 (0.005) | 0.084 (0.003) |
| Local reweighting | Sex | 0.781 (0.001) | 0.521 (0.03) | 0.323 (0.004) | 0.045 (0.005) | 0.08 (0.005) |

## C EVALUATION ON EXISTING FAIR FL

Table 3 presents the results for a fair FL algorithm proposed by Abay et al. (2020), which uses a reweighting algorithm proposed in the centralized setting (Kamiran and Calders, 2012). The algorithm involves assigning weights to each subgroup (defined by the label and a sensitive attribute) before FL, based on the local or global training dataset. This is called "local reweighting" or "global reweighting," respectively. During FL, parties update the FL model to minimize the weighted loss. The table presents the accuracy and fairness gap of the FL algorithm using local and global reweighting for the Income, Health, and Employment datasets.

We found that applying reweighing algorithms in FL reduces the average fairness gap across parties. This is due to the fact that, after reweighing, the parties that were initially biased do not introduce significant bias to the model during local training, resulting in a reduction of the fairness gap in the FL model.

However, we also noted that the fairness gap in the model remains high for the Race sensitive attribute, indicating that the global and local reweighing algorithms do not entirely eliminate bias in FL. Furthermore, we would like to highlight some concerns regarding the fair FL algorithm:

1. The fair FL algorithm enhances the performance of the minority group at the expense of the majority group. Figure 24 illustrates that the minority group (i.e., the Female group) has a better performance after the reweighing, but at the cost of reduced performance for the majority.

2. When improving fairness with respect to one sensitive attribute, the fairness issue with respect to another sensitive attribute may worsen, as shown in Table 3. For instance, when applying local reweighing to reduce bias with respect to the "Sex" attribute, the fairness gap with respect to "Race" increased from 0.514 (results on FedAvg) to 0.521, indicating that reweighing to improve fairness with respect to Sex may amplify the bias with respect to Race. This implies that the bias with respect to Race can still propagate in FL, and our analysis remains valid.

3. The reweighing algorithms assume that parties are interested in mitigating the bias with respect to the same sensitive attribute, which may not be the case in practice. Figure 25 shows the fairness gap of standalone models with respect to different sensitive attributes. We observe that many parties have a low fairness gap with respect to one sensitive attribute and a high fairness gap with respect to another sensitive attribute. Thus, parties may aim to enhance fairness with respect to different sensitive attributes, which renders this reweighing algorithm unsuitable. This problem is more prevalent in FL, given the different data distributions of parties.

Table 4: **Effect of FL on the real-world medical dataset - (ISIC2019).** The standard deviation across four runs is indicated between parentheses.

| Setting | Party 1 | | Party 2 | | Party 3 | | Party 4 | |
| Algorithm | Accuracy | $\Delta^{Acc}$ | Accuracy | $\Delta^{Acc}$ | Accuracy | $\Delta^{Acc}$ | Accuracy | $\Delta^{Acc}$ |
|---|---|---|---|---|---|---|---|---|
| Standalone | .658 (.008) | .024 (.018) | .972 (.009) | .013 (.007) | .77 (.024) | .06 (.04) | .717 (.057) | .019 (.006) |
| FedAvg | .546 (.117) | .056 (.044) | .908 (.118) | .047 (.058) | .702 (.108) | .13 (.025) | .536 (.108) | .066 (.027) |
| Centralized | .672 (.056) | .033 (.016) | .973 (.007) | .011 (.011) | .782 (.02) | .06 (.032) | .725 (.019) | .056 (.049) |

## D EXTENDING TO A REAL-WORLD MEDICAL DATASET

We evaluate the effect of FL on local fairness on a real-world medical dataset, ISIC2019 (Abay et al., 2020; Tschandl et al., 2018; Combalia et al., 2019), which contains dermoscopic images of skin lesions collected from six medical centers. The task is to classify dermoscopic images among nine different diagnostic categories. We filter out medical centers with less than 1900 images and regard each medical center with over 1900 images as a party (4 centers remain after filtering). For every party, we randomly select 1500 and 400 images for training and validation, respectively. We regard the binary notion of Sex as our sensitive attribute. In this multi-class and medical dataset setting, we measure the bias (fairness gap) based on the accuracy gap across groups. Following (Ogier du Terrail et al., 2022) [2], we train EfficientNets (Tan and Le, 2019), the state-of-the-art model structure for medical data, under the centralized, FL, and standalone setting. The accuracy and fairness gap for each client in different settings are shown in Table 4.

FedAvg does not improve accuracy, possibly due to data heterogeneity among parties (as shown in Figure 26), resulting in the model's instability and slow convergence, which is known as 'client-drift' issue (Karimireddy et al., 2020b).

Moreover, the FedAvg model results in a higher accuracy gap between groups, worsening the fairness issue compared to the standalone setting or the centralized setting (see the fairness gaps for party 1, party 3, and party 4 as examples). Our results suggest that, in real-world settings, FedAvg may not improve accuracy compared to the standalone setting for the local data distribution and may even exacerbate the fairness issue, leading to a more biased model.

---

[2]https://github.com/owkin/FLamby

# E  POTENTIAL MITIGATION

In this section, we suggest some potential solutions to reduce the effect of bias propagation in FL. Our hope is that this will inspire further research on developing fair FL in the future.

**Personalized FL**   Our study has revealed that aggregate updates, which involve the average local update from all parties, may result in a higher bias than local updates for less biased parties (see Figure 3). This can cause bias from other parties to propagate to the local model, thus increasing the local fairness gap. To mitigate the bias propagation effect, one possible approach is to avoid completely overwriting the local model with the global model. Personalized Federated Learning (PFL) techniques (Li et al., 2021) have recently been proposed to address this issue by focusing on improving the model's accuracy. PFL can be a promising direction for developing fairness-aware personalized federated learning approaches that can help reduce the bias propagation effect in FL.

**Fair Representation Learning**   Our analysis demonstrated that bias is encoded in a small number of parameters in the first layer of the neural network, which is typically considered the feature extractor (as illustrated in Figure 10). This suggests that the FL model is biased because the learned representation is biased, heavily influenced by the sensitive attribute. To address this issue, one possible solution is to implement fair representation learning techniques (Zemel et al., 2013; Zhao et al., 2019; Liu et al., 2022; Zeng et al., 2021b) that aim to learn a feature extractor which is fair, meaning that it minimizes the dependence on the sensitive attribute while still retaining enough information about the task (or input features). Therefore, using fair representation learning in FL can potentially mitigate the propagation of bias effects.

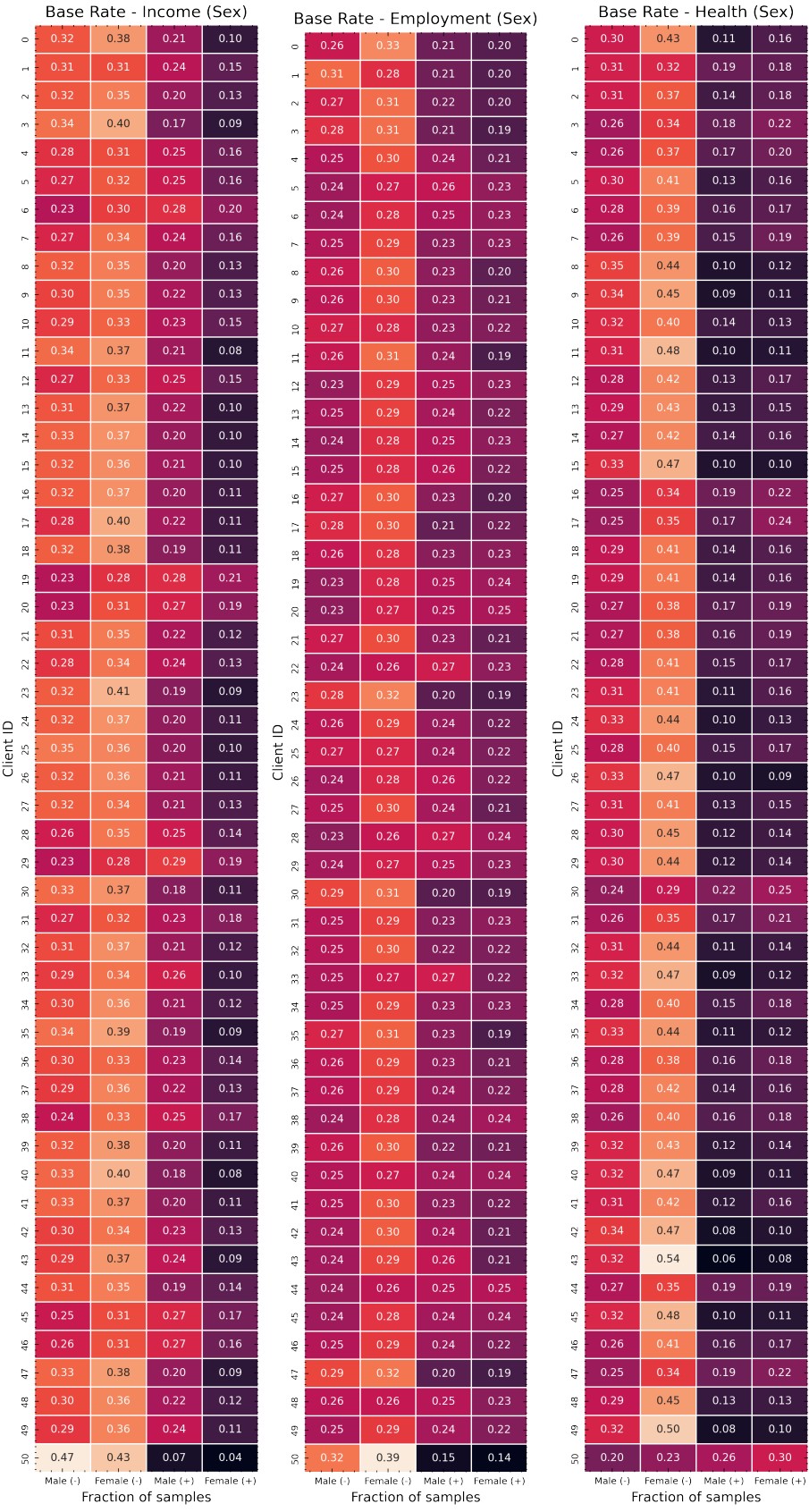

Figure 11: Fraction of samples for each subgroup - Sex

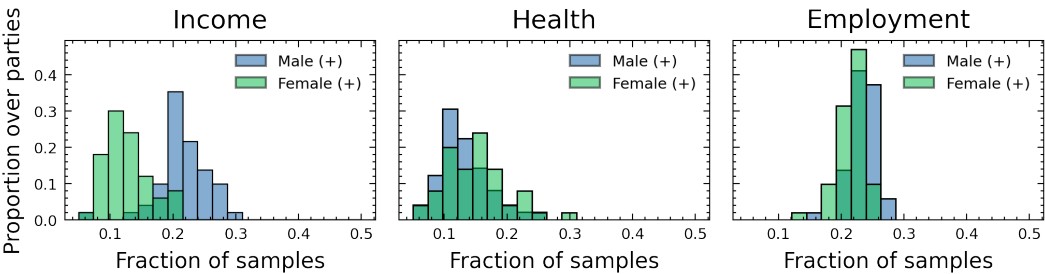

Figure 12: Histogram of the fraction of samples for each group with positive label - Sex

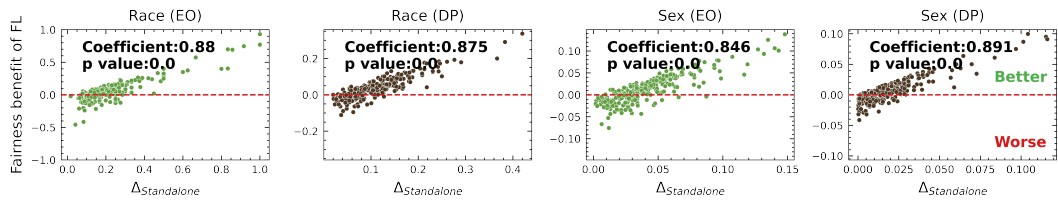

Figure 13: **Correlation between the fairness gap of the standalone model and the benefit obtained from FL - Health**

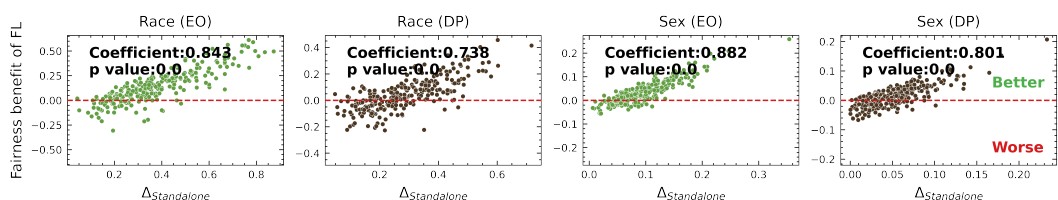

Figure 14: **Correlation between the fairness gap of the standalone model and the benefit obtained from FL - Employment**

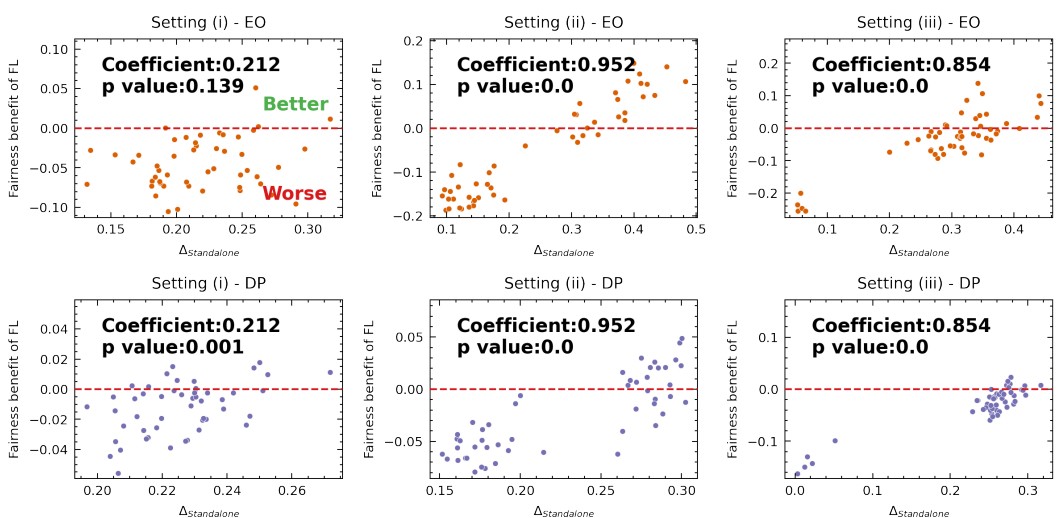

Figure 15: **Correlation between the fairness gap of the standalone model and the benefit obtained from FL - CelebA (Age)**

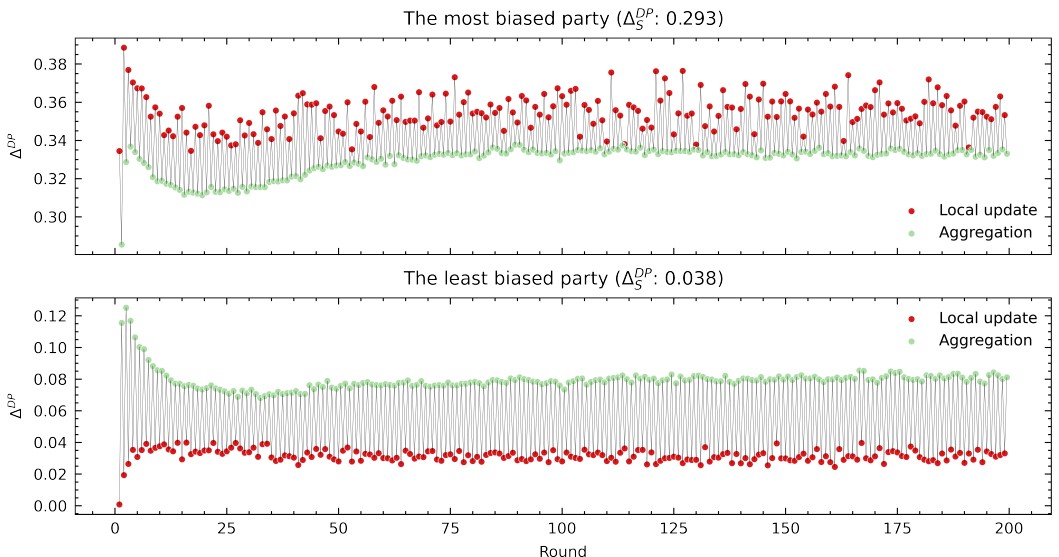

Figure 16: **Dynamic of $\Delta^{DP}$ during the training - Income (sex).** Figure shows the fairness gap for the aggregated model and local updated model from the most biased party and least biased party during the training. The most (least) biased party is the party with the highest (lowest) fairness gap in the standalone setting. The fairness gap in the standalone setting for those parties is shown in the title.

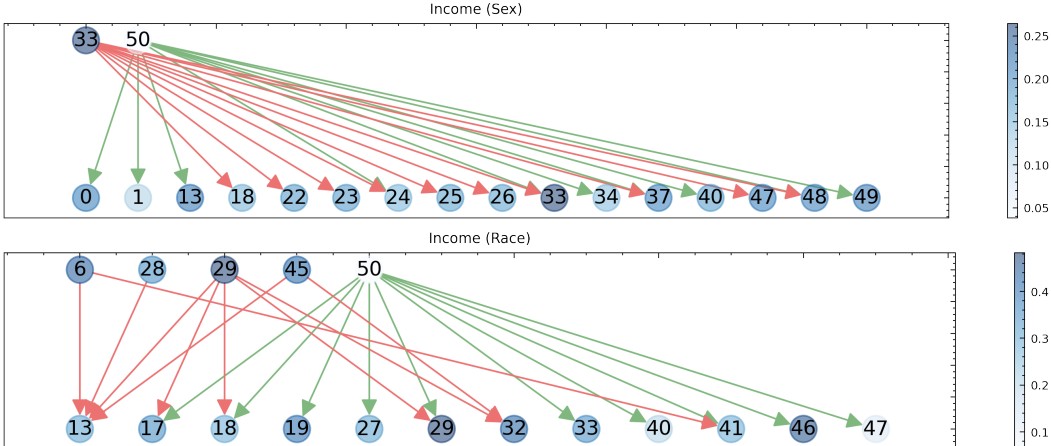

Figure 17: **Influence subgraph - Income** ($\Delta^{DP}$) Top 10 maximal positive influential pair and top 10 maximal negative influential pair. The green edge and red edge represent the positive and negative influence, respectively. The color of the node represents the fairness gap in the standalone setting.

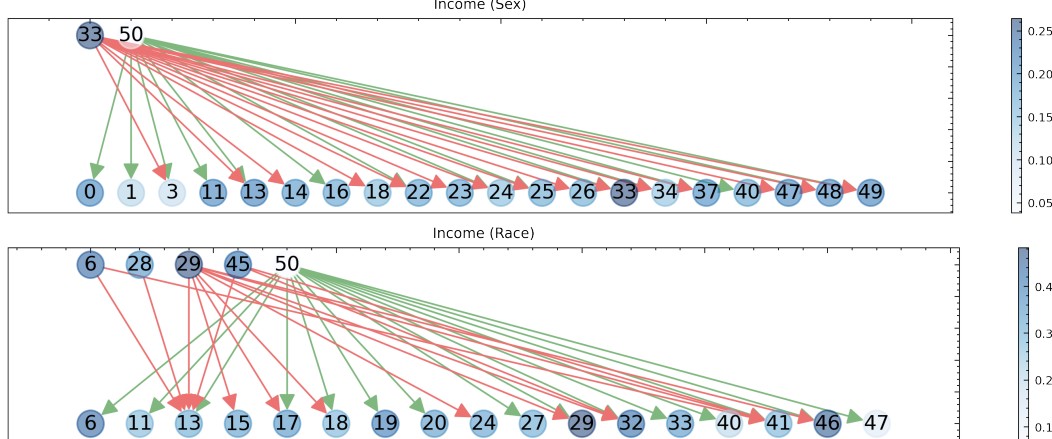

Figure 18: **Influence subgraph - Income** ($\Delta^{DP}$) Top 15 maximal positive influential pair and top 15 maximal negative influential pair. The green edge and red edge represent the positive and negative influence, respectively. The color of the node represents the fairness gap in the standalone setting.

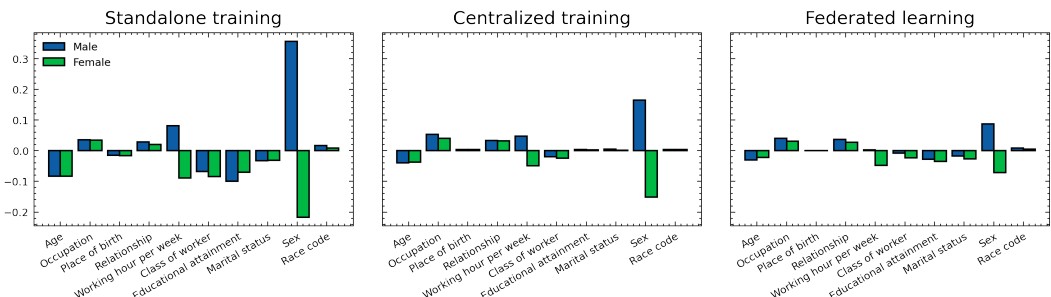

Figure 19: **Feature attribution value - Income (Most biased party)** We shows the feature attribution value for all the input features. The model trained in FL and standalone setting has a large dependency on the sensitive attribute "Sex".

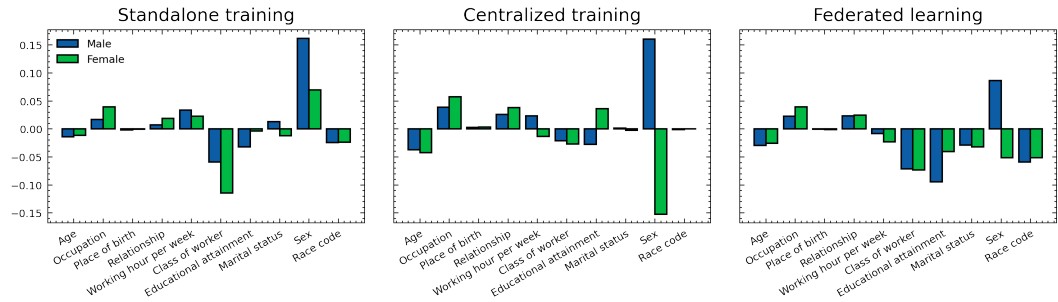

Figure 20: **Feature attribution value - Income (Least biased party)** We shows the feature attribution value for all the input features. The model trained in FL and standalone setting has a large dependency on the sensitive attribute "Sex".

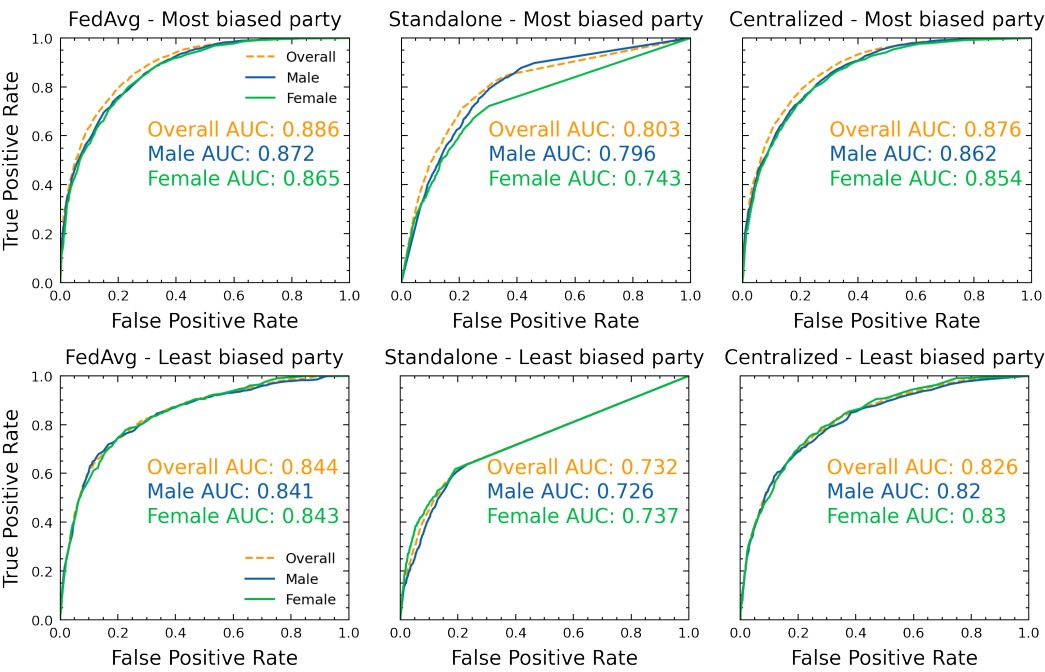

Figure 21: Receiver operating characteristic (ROC) of different models on groups - (Income, Sex)

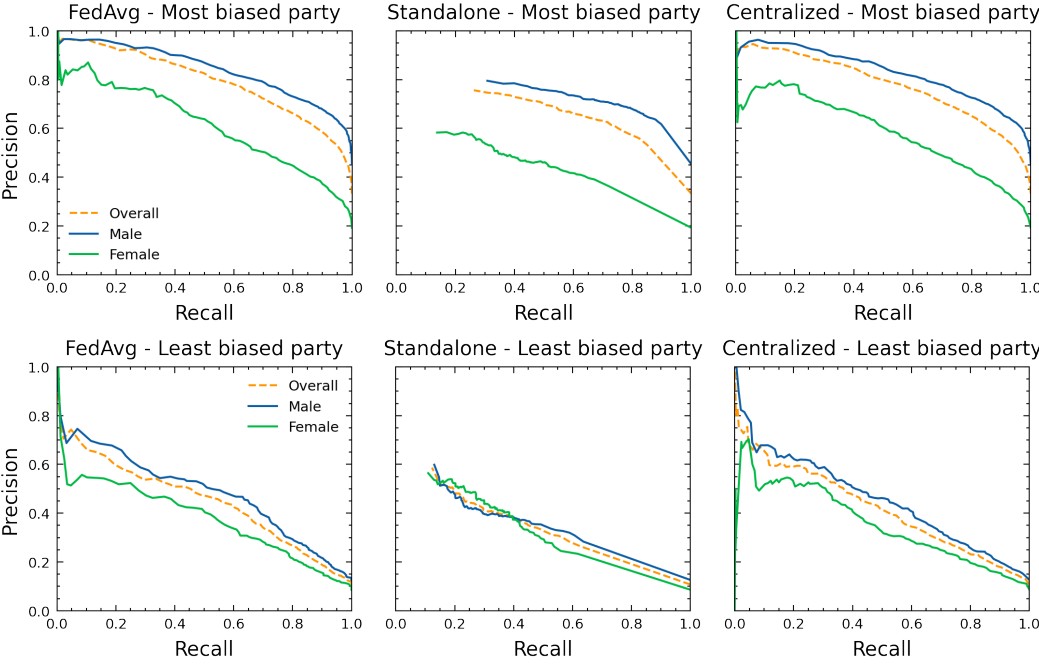

Figure 22: Precision-Recall curve (PRC) of different models on groups - (Income, Sex)

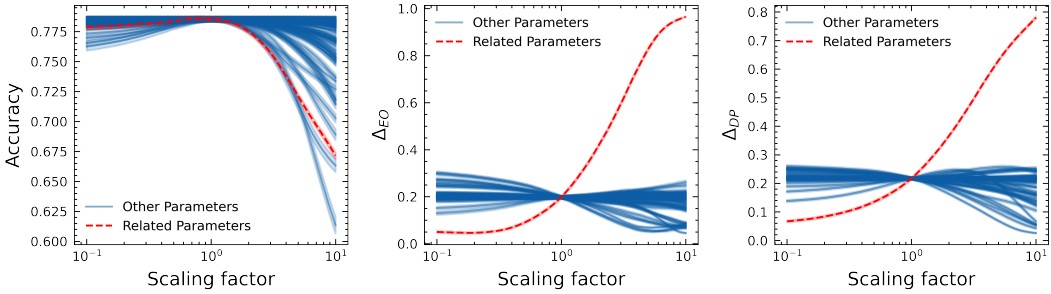

Figure 23: Effect of scaling model parameters - (Income, Sex)

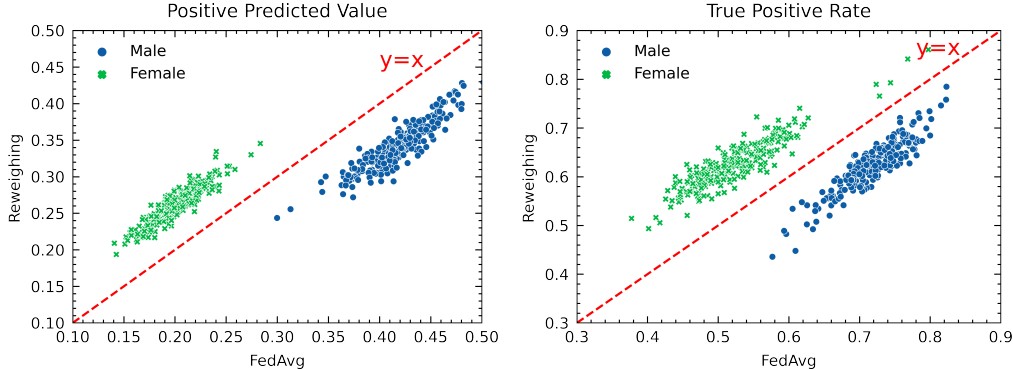

Figure 24: Effect of Fair FL algorithm (Abay et al., 2020) on group performance - (Income, Sex)

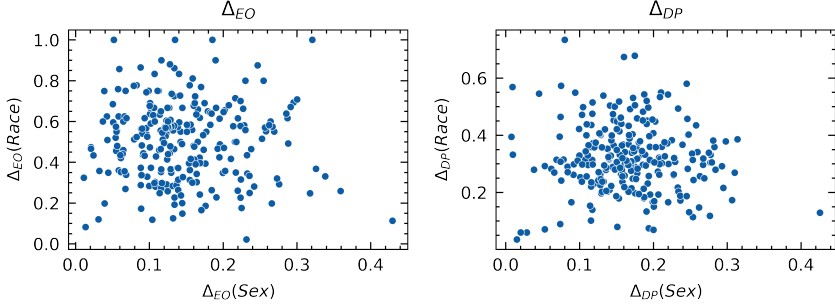

Figure 25: Fairness gap of standalone models with respect to different sensitive attributes - (Income). Each point represents the result for a single party in one run. We show the fairness gap with respect to different sensitive attributes differs a lot for each party.

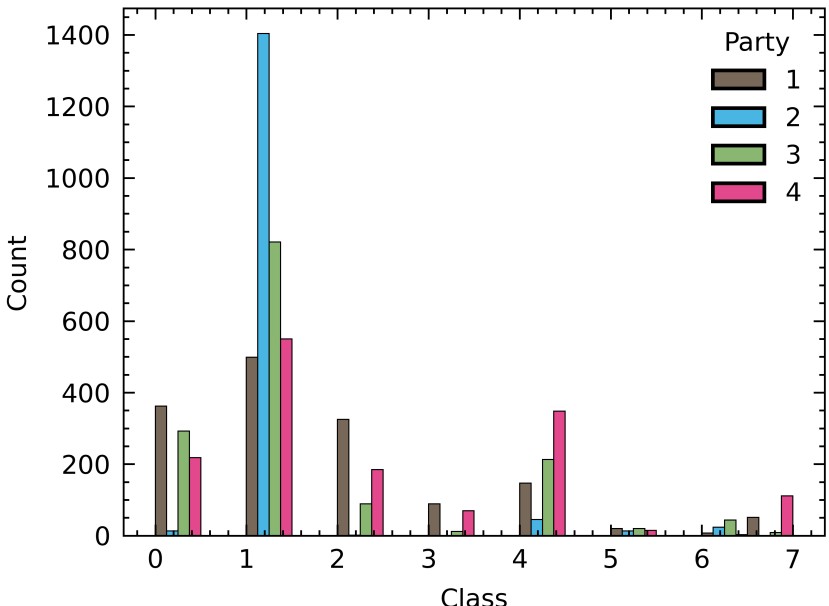

Figure 26: Number of samples from each class for each party - ISIC2019

