# OpenReview forum: "Bias Propagation in Federated Learning"
_ICLR.cc/2023/Conference — ICLR 2023 poster_

### Official Review · Reviewer_whH5 · 2022-10-21

**Confidence:** 2
**Correctness:** 3
**Technical Novelty And Significance:** 2
**Empirical Novelty And Significance:** 4
**Recommendation:** 6

**Clarity, Quality, Novelty And Reproducibility:**


In total, the authors study an interesting topic and present some useful insights, which can encourages more explorations about discerning the bias propagation in federated learning.

**Strength And Weaknesses:**

Strength

(1) It is a first attempt to systematically analyze the bias propagation problem in federated learning, which is in particular important considering the potential adverse impact of federated collaboration.

(2) A range of experiments have been conducted to understand how bias propagates among parties based on the metrics equalized odds and demographic parity. Specifically, several interesting insights have been discovered like "biased parties negatively influence other parties via aggregation throughout the training", and "large fairness gap is caused by disparate treatment".

Weakness

(1) Given the insights about the bias propagation, it will be better to discuss some potential solutions to avoid the bias propagation.

(2) The experiments that the bias is encoded in a few parameters are somewhat controversial, since it also possibly couples the heterogeneity and stochastic factors in the optimization.



**Summary Of The Paper:**

This paper studies the bias propagation in federated learning, which is an important issue in terms of the fairness. The authors conducted a range of experiments to understand the potential mechanism of bias propagation and present several interesting insights.

**Summary Of The Review:**

It is an interesting topic and I tend to accept this work, given the sufficient experimental evidence to understand the potential mechanisms of bias propagation in federated learning.

---

### Official Review · Reviewer_RzLQ · 2022-10-22

**Confidence:** 3
**Correctness:** 3
**Technical Novelty And Significance:** 2
**Empirical Novelty And Significance:** 2
**Recommendation:** 8

**Clarity, Quality, Novelty And Reproducibility:**

The authors do contribute a novel investigation of where bias in FL models might come from not just if biases exist. The presentation of the work was clear enough for me to understand. There are no obvious issues with quality and reproducibility.

**Details Of Ethics Concerns:**

This paper includes attractiveness prediction as a task. The paper is focused on evaluating the \textit{fairness} of attractiveness prediction but I am still not sure if we are still accepting attractiveness prediction as a task.


(Attractiveness task has been removed)

**Strength And Weaknesses:**

Strengths:

1. Authors go beyond observing disparity in FL and try to understand where model bias comes from.
1. Discovering that bias is encoded in a few parameters is very interesting. This seems to suggest a direction for mitigating bias.
1. Authors use an updated CPS dataset rather than UCI adult which is always a plus.

Weaknesses:
1. Demonstrations of disparities are dataset dependent and thus do not tell the most compelling story. Why don’t Health and Employment datasets experience increased bias due to federated averaging?
2. The authors use demographic parity as a fairness metric without discussing base rates. For example, It is unclear to me whether men and women have the same base rates in Celeb A. If they do not, then an increase in accuracy from a random classifier will necessarily imply a decrease in demographic parity.
3. Authors need to report p-values with (linear?) correlation coefficients
4. In 3.4, the authors point out that the large fairness gap is caused mainly by disparate treatment of the model on protected groups. Here we see that the sensitive attribute to be protected in this dataset is a key attribute for predicting the output both for the centralized model and for the FL model. In the real world, we can imagine using different thresholds for different groups to predict the correct outcome. In this case, overall and group-specific AUC and AUC disparity may give a clearer picture of what is going on.

Nit:
1. The language of “fairness benefit” vs “fairness gap” is confusing. My understanding is that fairness benefit is the difference between fairness gaps (Figure 2)
2. The authors mention heterogeneous distributions in the introductions but then do not discuss how heterogenous their datasets are.


**Summary Of The Paper:**

The authors focus on measuring bias in federated learning on tabular (US Census) and image  (Celeb A) datasets. They measure fairness using demographic parity and equalized odds. They find that:
1. Some federated averaging may increase the disparity in some datasets (Table 1)
2. FL improves fairness for stand-alone models with more disparity at the cost of worsening fairness for less biased parties (Figure 2)
3. Aggregation and local updates produce contradicting effects in fairness gap.
4. Less biased parties have a stronger positive influence in terms of fairness on other parties via aggregation. A similar effect follows with more biased parties and negative influence.
5. Model bias is caused by the models treating groups differently and the FL showing a more distinct separation in terms of sex feature attrition.
6. Biased parties use just a few parameters to encode bias through local updates and are propagated through aggregation.


**Summary Of The Review:**

The authors present several findings to measure biases in FL and investigate where they come from. There is nothing I strongly object to in this work but also nothing especially compelling. It is unclear what the take away message or the actionable insight is. I think the two main weaknesses are that most of the experiments are based on the Income dataset (the subset of the census that shows FL biases) and that there is no discussion about the potential unequal base rates between groups in these datasets. What would imposing demographic parity even achieve and is that desirable in an income dataset?

*** Updating review to accept after authors replaced attractiveness task ***

Future papers evaluating and addressing biases in FL should follow suit in avoiding attractiveness prediction as a task.

---

### Official Review · Reviewer_LgD7 · 2022-10-25

**Confidence:** 4
**Correctness:** 3
**Technical Novelty And Significance:** 2
**Empirical Novelty And Significance:** 3
**Recommendation:** 6

**Clarity, Quality, Novelty And Reproducibility:**

The paper is clearly written, and the quality is reasonable.


**Strength And Weaknesses:**

[Strength]

S1: The paper shows several good empirical observations, which further reveal the importance of fairness-aware federated learning.

S2: Also, the paper tries to give possible intuitions on why such unfair phenomena happen.

[Weakness]

W1: Although there are some interesting empirical results, the baselines are not enough to make the observations more reliable. Currently, the paper only focuses on one federated learning algorithm called FedAvg. Even though the paper writes the investigation on other algorithms is future work, it is currently hard to simply think that their results will be consistently observed in other standard federated learning algorithms. If the paper wants to focus on FedAvg, I would recommend clarifying the contribution limits of the paper.

W2: Also, the paper does not evaluate any previous fair federated learning algorithms in their setting. I understand that the assumptions on the data distribution are different from the previous works and this work, as most previous works assume a single global distribution and this work focuses on heterogeneous data distributions among parties. However, it will be still worth analyzing how other fair federated learning algorithms work in this paper’s setting. If the previous algorithms still mitigate the bias propagation in the heterogenous setting, then we can get more hints on how we can solve this issue. Such results will enhance the empirical study.

W3: Some of the observations are not very surprising. For example, the phenomenon that a more biased party negatively affects the fairness of other parties can be similarly observed in centralized learning (e.g., more biased subgroups will negatively affect the fairness of the entire model). Thus, it would be much helpful if the paper clarify why each observation is especially important in a federated learning setting compared to centralized training.


**Summary Of The Paper:**

The paper investigates unfairness phenomena in federated learning when parties have heterogeneous data distributions. Empirical results are gathered from the UCI Census and CelebA datasets. As a result, the paper observes several interesting patterns, including 1) the bias of each party differently affects model fairness (e.g., demographic parity and equalized odds) and 2) the bias may aggregate during the training. In addition, the paper gives some intuitions on why such a bias propagation occurs.

**Summary Of The Review:**

The empirical studies of this paper give several nice intuitions on fairness issues in federated learning on heterogenous data distributions. However, there are also some weak points that make the contribution of this paper less significant. Overall, I vote that this paper is marginally below the acceptance threshold.

========== Update after the rebuttal ==========

My major concerns have been mostly addressed, so I updated the score to 6.

---

### Author Response · Authors · 2023-02-09
**Revised to Meet Ethics Clearance Requirement**

Dear Ethics Committee,

We made the amendments required by the ethical review committee.

The initial ethical concern raised by Reviewer RzLQ about one of our many experiments is that "This paper includes attractiveness prediction as a task. The paper is focused on evaluating the \textit{fairness} of attractiveness prediction, but I am still not sure if we are still accepting attractiveness prediction as a task." (See [here](https://openreview.net/forum?id=V7CYzdruWdm&noteId=W_DVV0QCug)). The ethical committee also commented on the attractiveness prediction task that "continued use of someone else's data about attractiveness would merit some ethics commentary by the authors on the hazards of using beauty or attractiveness as any kind of legitimate factor at all." (See [here](https://openreview.net/forum?id=V7CYzdruWdm&noteId=dkLoKXBPuPh))

To address this concern, we promised to replace the results of the attractiveness prediction task with the age prediction task in the paper (See [here](https://openreview.net/forum?id=V7CYzdruWdm&noteId=5G1HVoeoKu)). This revision was successfully carried out, and **all references to the attractiveness attribute and attractiveness prediction task and its results have been completely removed from the paper**. In their place, we included the results of the age prediction task (in one row of Table 1 and Figure 13 in the Appendix). The results are consistent with those in the original version; thus, our overall analysis and conclusions remained unchanged.

We appreciate the opportunity to address this concern and stand ready to provide additional information if needed.

Paper5755 Authors

---

### Decision · Program_Chairs · 2023-01-20

**Decision:**

Accept: poster

**Justification For Why Not Higher Score:**

The results are interesting and the contributions are above the bar of acceptance but the significance of the results does not merit an oral or spotlight slot.

**Justification For Why Not Lower Score:**

The paper gives an insightful empirical study of an important phenomenon. The contributions are good enough to merit acceptance.

**Metareview: Summary, Strengths And Weaknesses:**

The paper investigates the challenges facing group fairness in federated learning (FL) and focuses on how bias against under-represented groups propagates to all parties in an FL algorithm. The authors conduct an extensive empirical study of this phenomenon on real-world datasets such as US census and CelebA datasets. The empirical results reveal several interesting findings, for example, they reveal that bias is encoded in a few model parameters and propagates to all parties via model aggregation, and that federated averaging can increase disparity in some datasets.

Understanding the notions of fairness and bias in federated learning is an important goal that could have profound impacts on society and on how federated learning algorithms are designed and deployed in practice. Thus, a careful and extensive study like the one conducted in this paper can be valuable and useful to the research community.

The authors have also done a good job addressing the reviewers' concerns and making necessary adjustments to improve their paper.

Also, initially there was a legitimate concern from one of the reviewers regarding the suitability of one of the task's considered by the authors in the CelebA dataset, but the authors replaced it during the discussion phase with another more suitable task. Hence, there is a consensus that this work is above the bar for acceptance.


**Note From Pc:**

if the above contains the word "oral" or "spotlight" please see: "oral" presentation means -> notable-top-5% and "spotlight" means -> notable-top-25%. As stated in our emails, we are disassociating presentation type from AC recommendations